# *GmIAA27* Encodes an AUX/IAA Protein Involved in Dwarfing and Multi-Branching in Soybean

**DOI:** 10.3390/ijms23158643

**Published:** 2022-08-03

**Authors:** Bohong Su, Haitao Wu, Yong Guo, Huawei Gao, Zhongyan Wei, Yuyang Zhao, Lijuan Qiu

**Affiliations:** 1College of Agriculture, Northeast Agricultural University, Harbin 150030, China; 18401609366@163.com (B.S.); wuhaitao0629@163.com (H.W.); 2National Key Facility for Crop Gene Resources and Genetic Improvement, Institute of Crop Science, Chinese Academy of Agricultural Sciences, Beijing 100081, China; guoyong@caas.cn (Y.G.); gaohuawei1025@126.com (H.G.); zhaoyuyang1428@163.com (Y.Z.); 3Institute of Plant Virology, Ningbo University, Ningbo 315211, China; weizhongyan@nbu.edu.cn

**Keywords:** soybean, plant height, branches, AUX/IAA, TIR1

## Abstract

Soybean plant height and branching affect plant architecture and yield potential in soybean. In this study, the mutant *dmbn* was obtained by treating the cultivar Zhongpin 661 with ethylmethane sulfonate. The *dmbn* mutant plants were shorter and more branched than the wild type. The genetic analysis showed that the mutant trait was controlled by a semi-dominant gene. The candidate gene was fine-mapped to a 91 kb interval on Chromosome 9 by combining BSA-seq and linkage analysis. Sequence analysis revealed that *Glyma.09g193000* encoding an Aux/IAA protein (GmIAA27) was mutated from C to T in the second exon of the coding region, resulting to amino acid substitution of proline to leucine. Overexpression of the mutant type of this gene in *Arabidopsis thaliana* inhibited apical dominance and promoted lateral branch development. Expression analysis of *GmIAA27* and auxin response genes revealed that some GH3 genes were induced. GmIAA27 relies on auxin to interact with TIR1, whereas Gmiaa27 cannot interact with TIR1 owing to the mutation in the degron motif. Identification of this unique gene that controls soybean plant height and branch development provides a basis for investigating the mechanisms regulating soybean plant architecture development.

## 1. Introduction

As a major grain and oil crop, soybean (*Glycine max* L.) provides high-quality protein and industrial raw material [1,2]. With the rapid increase in human consumption and industrial uses, the demand for soybean production has increased [3]. Plant architecture improvement is one of the effective methods for increasing soybean yield [4]. Soybean is a typical pod crop, and its yield components differ from those of cereal crops [5]. The plant architecture of soybean is unique, with the leaves, inflorescence, and pod all growing on nodes of the stem or branch. For this reason, soybean yield is strongly affected by the number of stem internodes and branches.

Plant height, the basis of plant architecture, is determined mainly by stem internode growth and directly affected by stem growth habits in soybeans [6]. An early study suggested that *Dt1* and *Dt2* are two major loci regulating soybean stem growth habit [6]. *Dt1* is a functionally conserved homolog gene of *Arabidopsis TERMINAL FLOWER1* (*TFL1*) and is a flowering-suppressor gene expressed mainly in shoot apical meristem (SAM) [7]. *Dt2* is a gene with a dominant MADS domain factor belonging to the APTALA1/SQUAMOSA (AP1/SQUA) subfamily. *Dt2* could reduce the expression of *Dt1* to promote the transition of shoot apical meristem from vegetative to reproductive growth, to induce flowering in the shoot apical tissue, and lead to transition from indeterminate to determinate stem growth habit [8]. Plant hormone synthesis and signal transduction are also involved in plant height formations in soybeans. *GmDW1* is a gene involved in gibberellin synthesis that regulates plant height in soybean and encodes an endoroot-kausenin synthase. A non-synonymous SNP mutation of the *GmDW1* gene prevents synthesis of bioactive GA_1_ and GA_4_, resulting in dwarf and shortened internodes. The normal plant height can be restored by external application of GA_3_ to the mutant [9]. Branching also affects soybean plant architecture. The regulation module of miR156-SPL plays a regulatory role in the process of branching development. Overexpression of *GmmiR156b* in soybean increased the numbers of nodes, pods, long branches, and stem diameter, but not affecting plant height, which could increase yield per plant by 46–63% by increasing 100-seed weight [10].

Auxin/indole-3-acetic acid (Aux/IAA) protein is a key factor in the response to auxin signal and functions in the regulation of auxin signal transduction, embryogenesis development, flowering, lateral root initiation and extension, hypocotyl elongation, and flower organ development [11,12]. AUX/IAA protein is a nucleoprotein characterized by four highly conserved domains (I–IV). Domain I acts as a transcription suppressor of auxin regulatory genes [13] and domain II is TIR1/AFB with a conserved denitrification sequence that recognizes GWPPV and regulates Aux/IAA protein stability [14]. Aux/IAA forms a heterodimer with Auxin Response Factors (ARFs), inhibiting the transcription activity of ARF at low auxin concentrations [15]. When auxin receptors (TIR1/AFBs) bind to auxin at high auxin concentrations, Aux/IAA proteins are degradation and ARFs are released from Aux/IAA to regulate downstream gene expression [14]. Domains III and IV can be homodimerized or heterodimerized with ARF to regulate the expression of auxin response genes *Aux/IAA*, *SAUR*, and *GH3* [16,17,18]. Gain-of-function mutations in domain II of Aux/IAA genes always result in inducing altered lateral root formation, apical dominance, stem elongation, leaf expansion, and gravitropism. For example, gain-of-function *axr3-1/iaa17-1* mutant showed shorter hypocotyls, upward-curling leaves, and no root hair [19].

In this study, a soybean mutant with dwarf and multi-branch number (*dmbn*) displaying an extreme reduction in plant height and multi-branching was identified, and the gene of *GmIAA27* was isolated by combining mapping-by-sequencing with linkage analysis. We performed gene function verification through transgenic experiments and expression analysis of *GmIAA27* and downstream *GH3* genes. Moreover, the protein interaction between GmIAA27 and GmTIR1d was verified by in vitro and in vivo to clarify the regulatory mechanism of GmIAA27 in plant height and branch development.

## 2. Results

### 2.1. Characterization of the dmbn Mutant

A *dwarf and multi-branch-number* soybean mutant named *dmbn* was identified from EMS-mutagenized seeds of Zhongpin661(wild type, Zp661) (Figure 1A). In comparison with the wild type, the mutant had short petioles and small involute leaves (Figure 1G). At the seedling stage, the mutant showed a strong ability to form lateral branch meristems, with the marked characteristic that lateral branches always originated at cotyledon nodes. Another characteristic was that the first trifoliate leaf grew opposite to the second (Figure 1F). The mutant was severely defective in lateral-root formation (Figure 1E). At maturity stage, it displayed a dwarf plant phenotype, and the branch number of the mutant was higher than that of the wild type (Figure 1B,I). Heterozygous plants were of moderate height, intermediate to the wild type and mutant. Heterozygous plants had more branches than either the mutant or the wild type (Figure 1D). The dwarfism phenotype of the mutant was due mainly to the reduction in internode number and internode length. Seeds of the mutant were uneven in size, with 100-seed weight of only 9.6 g. Thus, *dmbn* displayed a pleiotropic phenotype and reduced apical dominance, affecting most agronomic traits in soybean.

### 2.2. Cytological Observations

To study how the mutant’s defects in organ development are caused, cytological observations were performed by comparing the organs of the Zp661 and *dmbn* mutant. In comparison with the wild type, the axillary bud meristem of the mutant formed new lateral branches (Figure 2A,D). The major vein of the *dmbn* mutant was smaller than that of Zp661 (Figure 2B,E). By contrast, the vascular cylinders of the *dmbn* mutant, particularly the phloem, cambium, and pith, were smaller than those of Zp661(Figure 2C,F). The stem of the mutant was thinner and softer than that of the wild type, mainly because the number and size of cell layers of sacroma, parenchyma, conduit, and pith were smaller than those of the wild type (Figure 2G–J).

### 2.3. Mapping of dmbn Gene with BSA-Seq

In the selfed population of multiple heterozygous lines, 20 mutant phenotypes and 20 wild-type plants were selected to form mutant and wild-type pools. The raw data of 83.76 Gbp were obtained from the two pools by whole-genome sequencing. After removal of low-quality reads and adapter contaminants, WT, and mutant pools retained 82.93 Gbp of clean reads. The proportion of clean reads with Q30 scores was 90.62%, and the mean GC content was 35.68% (Appendix A). The average coverage depth was 38.07× and the average percentage between sample and reference genome was 99.22% (Appendix A). Genome coverage to a depth of at least one base was 99.41% (Appendix A).

Compared with the number of SNPs in the reference genome, 252,209 SNPs and 253,250 SNPs were identified in the mutant and wild type pools separately. The number of SNPs common to both the mutant and wild type pools was 243,989 (Appendix A). For association analysis, 261,470 SNPs were selected, those supported by fewer than 4 reads. Finally, 135,970 reliable SNPs with high quality were obtained for BSA-seq mapping.

To identify the candidate gene controlling the variation phenotype of the *dmbn* mutant, the Euclidean distance (ED) algorithm was used to obtain the genetic distance to the associated genes. All ED values were then fitted and plotted across each chromosome interval. The threshold ED value for all SNPs was calculated as 0.05. Seven significant peaks including nine intervals were identified in SNP-ED plotting, which might be candidate region for regulating dwarf and multi-branching (Appendix A). The sizes of these intervals varied from 140 Kb to 5.07 Mb, with a total length of 8.47 Mb spread across three chromosomes. The highest peak was located on chromosome 9 at physical position 39.78–44.85 Mb which is the candidate interval (Appendix A). A total of 1546 genes were contained in the nine candidate intervals, only three of which carried nonsynonymous mutation SNPs (*Glyma.09G189300*, *Glyma.09G193000*, *Glyma.09G199500*).

### 2.4. Fine Mapping of the GmIAA27 Gene

To further narrow the candidate interval in chromosome 9, a genetically isolated population containing 185 F_2_ plants was obtained by crossing a soybean cultivar Dengke1 (DK1) with the *dmbn* mutant. F_2_ population displayed segregation of 48 wild-type, 42 mutant-type, and 94 intermediate-type plants. The segregation fits to a 1:2:1 ratio (χ^2^ = 0.28 < χ^2^_0.05_ = 3.84), suggested that the phenotype of the *dmbn* mutant was controlled by a single semi-dominant locus. SSR markers were selected from the candidate interval of chromosome 9 for polymorphism screening and two SSR markers (09-1257, 09-1440) linked to the target loci were identified (Figure 3A). In total, 185 F_2_ plants were genotyped using these two linked markers, and 42 recombinants were identified. A total of 11 additional polymorphic markers were found between the two markers and used to genotype recombinants. Additionally, 4 recombinant plants (1-6, 2-44, 2-89, 3-32) were identified and the candidate interval was narrowed to a segment with a physical size of approximately 91 kb between markers 09-1277 and 09-1285 (Figure 3B). There were 10 predictive genes in the fine mapping region, among them only 1 gene (*Glyma.09G193000*) with a nonsynonymous mutation (Figure 3C). *Glyma.09G193000* contains five exons and four introns with a 1739-bp transcript and encodes an Aux/IAA protein of 308 amino acids. The single-nucleotide transition (C to T) caused a substitution of proline (Pro) by leucine (Leu) at position 160 (designated as P160L) in the degron motif (GWPPV) of conserved domain II (Figure 3D). *Glyma.09G193000* is a homolog of *Arabidopsis IAA27* and we accordingly assigned *Glyma.09G193000* as the candidate gene for *dmbn* and named it *GmIAA27*. Sequence alignment of GmIAA27 showed that GmIAA27 had conserved amino acid with other homologue genes in motif (GWPPV) around 160th amino acid (Figure 4A,B). Phylogenetic analysis revealed that GmIAA27 and two other soybean genes (*Glyma.05G229300* and *Glyma.08G036400*) has high similarity with *Arabidopsis* IAA27 (Figure 4C).

### 2.5. Functional Confirmation by Arabidopsis Transformation

To verify that *GmIAA27* was the target gene regulating *dmbn* phenotype, the *GmIAA27* and *Gmiaa27* was ectopically expressed under the control of the 35 S promoter in *Arabidopsis* ecotype Columbia 0 (Col-0). The morphology of transgenic plants expressing the *GmIAA27* allele was identical to that of the wild type. In contrast, transgenic plants expressing the *Gmiaa27* allele displayed small and curly leaves at the seeding stage and a marked dwarf stature without apical dominance at maturity (Figure 5A). Moreover, transgenic plants overexpressing *Gmiaa27* had small and short siliques with low seed setting rate (Figure 5B).

### 2.6. Subcellular Localization of GmIAA27 and Sequence Analysis

To determine the subcellular location of GmIAA27 and the mutated Gmiaa27 protein, vectors for the GmIAA27-GFP fusion protein and mutated Gmiaa27-GFP fusion protein were transiently expressed in *Nicotiana benthamiana* leaves and onion epidermal cells, respectively. Vectors for the single GFP were transiently expressed as a control. Expression of the GmIAA27/iaa27-GFP fusion gene under the control of the 35 S promoter in the *N. benthamiana* leaves and onion epidermal cells was expressed mainly in the nucleus and cytoplasm (Figure 6).

### 2.7. Quantitative Real-Time PCR Validation of GmIAA27 Genes

To investigate the biological function of *GmIAA27*, quantitative PCR was used to study the expression patterns *GmIAA27* gene in roots, stems, and leaves. The results showed that the expression level of *GmIAA27* with higher transcript abundance in stem, and lower in root (Figure 7A). The auxin/indoleacetic acid (Aux/IAA), the GH3, and the *Small Auxin-Up RNA* (SAUR) gene families are three major classes of auxin-responsive genes [20]. Previous results showed that point mutations in the degron domain of AUX/IAA often regulate the expression of the downstream GH3 gene family by binding to ARFs [21]. To test whether the expression level of GH3 gene is affected in the mutant, the relative expression of GH3s genes among in several organs of *dmbn* and Zp661 plants were compared by qRT-PCR. In contrast with Zp661, four genes (*Glyma.05G101300*, *Glyma.12G103500*, *Glyma.03G149400*, and *Glyma.04G006900*) were significantly up-regulated in *dmbn* leaf tissues (Figure 7B), five (*Glyma.17G165300*, *Glyma.01G190600*, *Glyma.12G103500*, *Glyma.06G260800*, and *Glyma.03G149400*) in *dmbn* stem tissues (Figure 7C), and five (*Glyma.17G165300*, *Glyma.01G190600*, *Glyma.05G101300*, *Glyma.12G103500*, and *Glyma.06G260800*) in *dmbn* root tissues (Figure 7D).

### 2.8. Mutation of GmIAA27 Affects Its Interactions with GmTIR1 under NAA Treatment

According to the Glyma.Wm82.a2.v1 reference genome, four GmTIR1(GmTIR1a-d) were used to investigate interactions with GmIAA27 using yeast two-hybrid (Y2H) assays. The results revealed that GmIAA27 interacted with GmTIR1d with 100 μM NAA treatment, but not interacted without NAA. However, Gmiaa7 did not interact with GmTIR1d either with or without auxin treatment. We reserved a region of the GmIAA27 amino acid sequence containing the degron motif GWPPV for Y2H, which further verified the P160L substitution in the degron motif GWPPV that abolished the interaction between Gmiaa27 and GmTIR1d (Figure 8A). The interaction between GmIAA27 and GmTIR1d under NAA treatment was further confirmed by pull-down assay in vitro (Figure 8C) and co-immunoprecipitation (Co-IP) assay in vivo (Figure 8B).

## 3. Discussion

Plant height and branch number are critical determinants of soybean plant architecture and yield. In this study, several aspects of the *dmbn* mutant phenotype that include extreme dwarf plant height (35 ± 0.5 cm) with a thin stem and more branches. This phenotype closely resembles the description of the rapeseed mutant *sca*, which has semi-dwarf stature [21]. The *dmbn* mutant also displayed a short petiole and small involute leaves, phenotypes similar to those of gain-of-function mutations in *Arabidopsis thaliana*, *Lycopersicon esculentum,* and *B. napus* Aux/IAA genes [22,23,24]. The mutant, with this special phenotype, provides a material basis for investigating the molecular mechanisms of soybean dwarfing and branch development.

With the development of next-generation sequencing (NGS) technology, it has shown great advantages in the identification of candidate genes among plant species [25]. Bulk-segregant analysis (BSA) can be used to detect and annotate loci associated with target traits and to study the genes that control the target traits and their molecular mechanisms. According to the parental origin of the target traits, BSA-seq can be divided into two directions: extreme traits (QTL-seq) and artificially mutagenized traits (MutMap) [26,27]. As an upgraded method of MutMap, MutMap+ uses mainly mixed samples composed of individuals with extreme phenotypes in a segregating population of selfed heterozygous plants [28]. In the process of gene mapping, BSA-seq can quickly reveal candidate intervals of target traits and more candidate genes, and in plants with high genetic transformation efficiency, such as rice or *Arabidopsis thaliana*, functional verification can be performed by genetic transformation to identify the target gene [29,30]. However, for soybean, a crop that is difficult to genetically transform, identifying a target gene in a large candidate interval is laborious and time-consuming [31,32]. In the present study, a mapping strategy combining MutMap+ methods and linkage analysis was adopted to discover candidate genes and reduce eight false-positive candidate intervals [33,34,35]. Accordingly, when we used linkage analysis to confirm the candidate interval, the candidate interval was anchored to chromosome 9 very quickly, greatly reducing the workload of screening candidate genes. Ultimately, the *GmIAA27* gene was identified by analysis of SNP data from the two bulks. Accelerated identification of candidate gene used a strategy that combines BSA-seq and conventional mapping methods.

Most IAA family genes have gain-of-function mutations in rice, Arabidopsis, and Brassica napus. Therefore, mutants with corresponding phenotypes will be generated after ectopic expression or ontology expression, for example, *BnaC05.iaa7* driven by their native promoters to transform WT Arabidopsis displayed curly and wrinkled leaves at the seedling stage and an obvious dwarf stature without apical dominance at maturity [24]. In this study, the mutant *Gm*iaa27 was transferred into WT Arabidopsis that showed the same phenotypic, including deformed leaf and without apical dominance, which fully indicated that the *GmIAA27* gene directly affected the growth and development of the plants. AUXIN/INDOLE-3-ACETIC ACID (Aux/IAA) proteins share a conserved domain structure and are usually localized in the nucleus, for example, AtIAA26, AtIAA17, ZmIAA2, ZmIAA11, and ZmIAA15 are confined to the nucleus, while some AUX/IAA are localized in both the nucleus and the cytoplasm, including ZmIAA20, ZmIAA33, and AtIAA8. LSD1 (LESIONS SIMULATING DISEASE RESISTANCE 1) negatively controls cell death and disease resistance which interact with IAA8 [36,37]. Therefore, GmIAA27 might not act exclusively on transcriptional regulation in the nucleus, but may also play a role in cytoplasmic processes via an unknown mechanism.

Aux/IAA is a plant-specific transcription regulator and a key component of the IAA signal transduction pathway, which regulates plant growth and development [38]. *GmIAA27* encodes an auxin-responsive protein (IAA). Usually, AUX/IAA protein is a repressor, which inhibits the regulation of downstream genes by ARF by forming a dimer with ARF [39]. After binding to TIR1 and sensing high concentrations of auxin, AUX/IAA undergoes ubiquitination degradation [40]. Domain II in AUX/IAA is a key domain to maintaining protein stability. The amino acid substitution will make it unable to interact with TIR1, and AUX/IAA is difficult to degrade [41]. Examples are the G84E substitution in the *BnaA3.IAA7* gene [21] and the P87L substitution in the *BnaC05.IAA7* gene [24]. In this study, the degron domain of GmIAA27 exhibited highly conserved properties (Figure 4A,B). In addition, we found that expression of Gmiaa27 always resulted in decreased protein accumulation in tobacco, this phenomenon still needs further study. Although the expression level of Gmiaa27 was low in tobacco leaves, the Co-IP interaction results similar to pulldown and Y2H were also detected after enrichment of Gmiaa27 protein. So, we speculate that the replacement of the first P by L in the degron motif GWPPV of *GmIAA27* led directly to the weakening or absent the interaction between Gmiaa27 and TIR1d even in the presence of auxin [37,42].

In the auxin pathway, Aux/IAA proteins activate downstream genes by inhibiting function of the ARFs that act as negative factors for target gene expression, via protein–protein interaction [43]. ARFs are a family of plant-specific DNA-binding proteins that bind to auxin-responsive promoter elements (AuxREs), which are present in auxin-responsive early genes to promote or repress transcription by *Aux/IAA*, *SAUR*, and *GH3* [18,44]. *DFL1* is a member of the GH3 family, and dfl1-D and DFL1 sense and antisense transgenic plants showed different lateral root and shoot phenotypes each other, caused by the expression level change of the *DFL1* gene [45]. A GH3 gene named *YDK1*, when overexpressed in wild-type *Arabidopsis*, confers small, epinastic leaves, short stems, and reduced apical dominance [46]. In the present study, the expression level of the *GH3* gene was significantly increased in multiple tissues, and different *GH3* genes were highly expressed in different tissues, and the expression changes of these genes may have been regulated by ARF. Because we identified the AuxREs element form upstream of the start codon of *GH3* genes (data not shown), we speculated that the high expression of these genes was responsible for the phenotypic changes.

## 4. Materials and Methods

### 4.1. Plant Materials and Growth Conditions

The soybean cultivars Zhongpin 661 (Zp661), and Dengke 1 were obtained from the Institute of Crop Science, Chinese Academy of Agricultural Sciences. The *dmbn* mutants were created by ethylmethane sulfonate mutation of Zhongpin 661 [47]. The *dmbn* mutant and Dengke 1 were crossed and the F_1_ were selfed to generate an F_2_ population for genetic analysis and mapping. Wild-type *Arabidopsis* seeds were surface sterilized with 75% ethyl alcohol for 10 min, flushed with sterile water three times, and transferred to half-strength Murashige and Skoog (1/2 MS) medium and soil for growth at 22 °C under conditions of 16 h light, 8 h dark, and 75% humidity.

### 4.2. DNA Extraction and Genetic Mapping

In total, 20 plants with extreme phenotypes were selected from different lines of heterozygous mutants. Genomic DNA was extracted by the modified CTAB method [48]. The DNA of each plant was equally mixed into two extreme pools for BSA-seq. To narrow the localization interval, polymorphic SSR markers on chromosome 9 were identified in the two parents [49]. The F_2_ plants were genotyped with these markers.

### 4.3. RNA Extraction and Real-Time Quantitative PCR

Total RNA was isolated from roots, leaves, and stems of soybean at the V1 stage with an RNA prep pure plant kit (Cat. No. R013-50, GeneBetter, Beijing, China). First-strand cDNA was reverse transcribed with a PrimeScriptTM RT reagent kit (Cat. No. RR047A. TaKaRa). qRT-PCR was performed according to the manufacturer’s instructions (Cat. No. MF013-01, Mei5 Biotechnology Co., Ltd., Beijing, China) on an ABI 7300 PCR system. All reactions were performed with three biological repeats. The soybean *Actin11* gene (*Glyma.18G290800*) was used as an internal control [50].

### 4.4. Cytological Observation

Leaves, stems, and axillary buds of Zp661 and *dmbn* mutants were fixed overnight in 15% FAA fixative solution at 4 °C. They were then dehydrated in an ethanol series of 75%, 80%, 90%, and 100% for 0.5 h per step, then immersed in xylene: ethanol mixtures of 50:50, 75:25, and 100:0 for 0.5 h per step, and finally immersed in paraffin: xylene mixtures of 25:75, 50:50, 75:25, and 100:0 for 4 h per step. The samples were embedded in paraffin, trimmed to appropriate shapes with a knife, and sliced with a hand-cranked microtome. The sections were placed on glass slides and incubated at 65 °C for 1 min and then at 37 °C overnight. They were then immersed twice in 100% xylene for deparaffinization for 20 min each time, followed by immersion in xylene: ethanol 50:50 for 5 min and then in 100%, 95%, 80%, and 70% ethanol for 2 min each. The sections were stained with safranin and fast green, sealed with neutral gum, air-dried, and observed and photographed under a microscope.

### 4.5. Subcellular Localization

Homologous recombination primers were designed for vector construction according to the CDS of GmIAA27/iaa27. The Zp661 and mutant cDNAs were used as templates for amplification with high-fidelity enzymes, and the target bands were recovered using the kit. The resulting target fragments were inserted into the pCAMBIA1305-GFP vector digested with *Xba*I and *Bam*HI, respectively, and were named pCAMBIA1305-GmIAA27/iaa27-GFP. The vectors were confirmed by sequencing. The plasmid was transferred into *Agrobacterium* strain EHA105 by electroporation, and single clones were picked for culture and bacteria preservation. PCR positive detection was performed at the same time. After activation of P19 bacterial solution and target *Agrobacterium* solution, 250 μL were pipetted into 5 mL of LB containing cannabidiol and rifampicin resistance and cultured to an OD_600_ of 0.8–1.0. To a new 2 mL centrifuge tube was added P19 bacterial solution and the target bacterial solution to a final concentration of P19 of 0.1 and a target bacterial solution concentration of 0.2, the solution was centrifuged at 4000× *g* for 5 min and the *Agrobacterium* was collected, twice resuspended in 1 mL of induction medium, and centrifuged again at 4000× *g* for 5 min. Finally, 2 mL of induction medium was added to resuspend and protected from light for 4 h before injection into tobacco. The injected tobacco was cultivated for 2 days for laser confocal observation.

### 4.6. Multiple-Sequence Alignment

The amino acid sequence of the *GmIAA27* gene was searched against the phytozome database using the BLASTP function, and the amino acid sequences of similar genes of soybean, rice, and *Arabidopsis* were retrieved. Multiple alignments were performed with DNAMAN software. WebLogo (http://weblogo.berkeley.edu/; accessed on 12 May 2022) was used to draw a conserved logo map.

### 4.7. Genetic Transformation of Arabidopsis

Full-length coding sequences of *GmIAA27/iaa27* were cloned into the pBI121 vector, which was named pBI121-GmIAA27/iaa27. The plasmid was introduced into *Agrobacterium tumefaciens* strain GV3101 for *Arabidopsis* transformation using the floral dip method [51].

### 4.8. In Vitro Pulldown Assay

Full-length coding sequences of *GmTIR1d* were cloned into the pCAMBIA1300-flag vector, which was named pCAMBIA1300-GmTIR1d-flag and introduced into the *Agrobacterium tumefaciens* strain EHA105 for expressing the fusion protein from *N. benthamiana* leaves. Full-length coding sequences of GmIAA27/iaa27 were cloned into the pGEX4T-1 vector and introduced into *E. coli* strain *BL21(DE3)* with 0.4 mM isopropyl-β-D-thiogalactoside (IPTG) at 28 °C for 16 h for expressing the fusion protein. For pulldown assay, after the fusion protein was purified with GST magnetic beads, the GmTIR1-Flag fusion protein was added with absence or presence of 100 μM NAA (1-naphthlcetic acid), and a protease inhibitor cocktail was added, followed by incubation at 4 °C overnight and the beads washing three times with 1× PBS. To the beads was added 2× protein loading buffer followed by denaturing at 100 °C for 10 min and then separation on 10% SDS-PAGE gel. GST fusion protein was detected with anti-GST (1:5000 dilution; MBL, PM013-7), and the Flag fusion protein was detected with anti-Flag (1:2000 dilution; MBL, PM185-7).

### 4.9. Co-Immunoprecipitation

pCAMBIA1300-GmTIR1d-Flag and pCAMBIA1305-GmIAA27/iaa27-GFP *Agrobacterium* were used to transfect *N. benthamiana* leaves, as described previously [52]. GFP-Trap (Chromotek, gtma-20) was used to immuno-precipitate the GmIAA27 complexes, followed by detection with anti-Flag (1:2000 dilution; MBL, PM185-7) and anti-GFP) (1:5000 dilution; MBL, 598-7) monoclonal antibodies.

### 4.10. Yeast Two-Hybrid Assay

Full-length coding sequences of *GmTIR1a* (Glyma.02G152800), *GmTIR1b* (Glyma.10G021500), *GmTIR1c* (Glyma.19G206800), and *GmTIR1d* (Glyma.03G209400) were cloned into the pGADT7 vector, and GmIAA27/iaa27 and a partial amino acid sequence of GmIAA27/iaa27(140–180) were cloned into the pGBKT7 vector. The plasmid combination was introduced into strain AH109 of *E.coli* and tested according to the manufacturer’s instructions (Clontech). Empty vectors were used as negative controls. Transformed yeast was cultured on plates containing SD/-Ade/-His/-Leu/-Trp medium with absence or presence of 100 μM NAA to test for interactions between the AD and BD fusion proteins.

## 5. Conclusions

We identified the phenotypic characteristics of a dwarf and multi-branched soybean mutant, and investigated its regulatory mechanism. The *dmbn* mutant was defective in plant height and leaf development and had the characteristics of multi-branching. The development of vascular bundle cells in leaves, cortex, sieve tube, and the pith cells in the stem were affected. The causal gene *GmIAA27* gene was isolated by BSA-seq and linkage analysis and verified through genetic transformation. The interaction between GmIAA27 and GmTIR1d requires the presence of auxin and affects the expression of downstream auxin-responsive genes involved in soybean growth and development. In summary, our study identified a gene that blocks auxin signaling and affects plant height and branch development in soybean. This finding provides a basis for studying the mechanisms regulating soybean plant architecture.

## Figures and Tables

**Figure 1 ijms-23-08643-f001:**
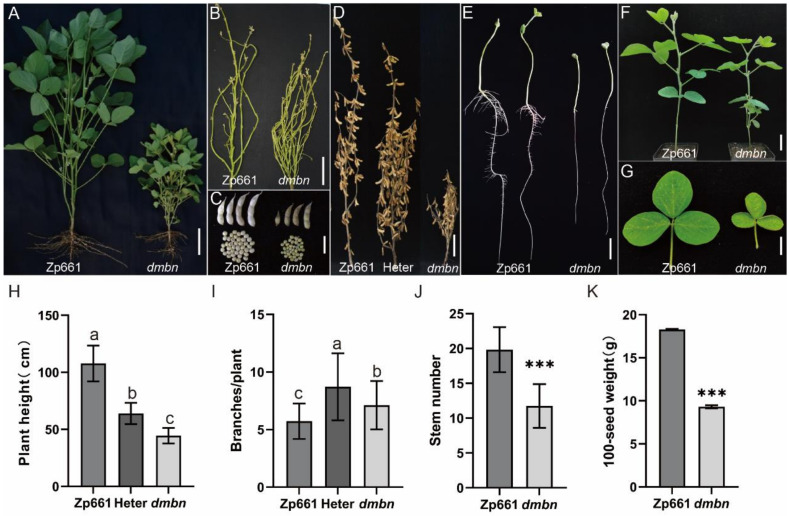
Phenotypic characterization of the *dmbn* mutant and Zhongpin661 (Zp661). (**A**,**B**) Phenotypes of Zp661 and *dmbn* at vegetative stage; bar, 10 cm. (**C**) Pods and seeds of mature soybean of Zp661 and *dmbn*; bar, 3 cm. (**D**) Phenotypes of Zp661, *dmbn*, and a heterozygous plant at the mature stage; bar, 10 cm. (**E**) Root phenotypes of Zp661 and *dmbn* at the seeding stage; bar, 3 cm. (**F**) Phenotypes of Zp661 and *dmbn* at the R5 stage; bar, 3 cm. (**G**) Leaf phenotypes at the R5 stage of Zp661 and *dmbn*; bar, 3 cm. (**H**,**I**) Plant height and branches of Zp661, *dmbn*, and a heterozygous plant at maturity. Values are mean ± SD (*n* = 15). (**J**,**K**) Stem number and 100-seed weight of Zp661 and *dmbn*. Student’s *t*-test was used for comparisons (*** *p* ≤ 0.001).

**Figure 2 ijms-23-08643-f002:**
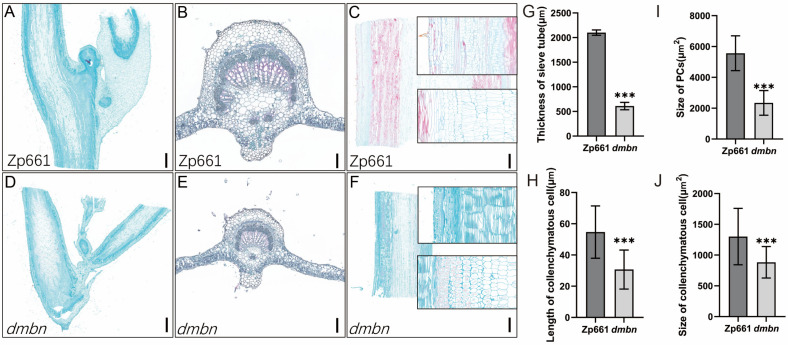
Cell elongation and expansion are defective in Zp661 and the *dmbn* mutant. (**A**,**D**) Axillary bud cross sections of Zp661 (**A**) and *dmbn* (**D**) from plants at the R5 stage; bar, 1000 µm. (**B**,**E**) Main vein cross sections of Zp661 (**B**) and *dmbn* (**E**) from plants at the R5 stage; bar, 200 µm. (**C**,**F**) Stem longitudinal sections of Zp661 (**C**) and *dmbn* (**F**) from plants at the R5 stage; bar, 1000 µm. (**G**–**J**) Statistical analysis of the sieve tube, collenchymatous cell and PCs (pith cells). Error bars ± SD (5 and 6 cells in Zp661 and *dmbn* of thickness of sieve tube; 65 and 60 cells in Zp661 and *dmbn* of pith cell length or size; 94 and 74 cells in Zp661 and *dmbn* of collenchymatous cell length. Student’s *t*-test was used for comparisons (*** *p* ≤ 0.001).

**Figure 3 ijms-23-08643-f003:**
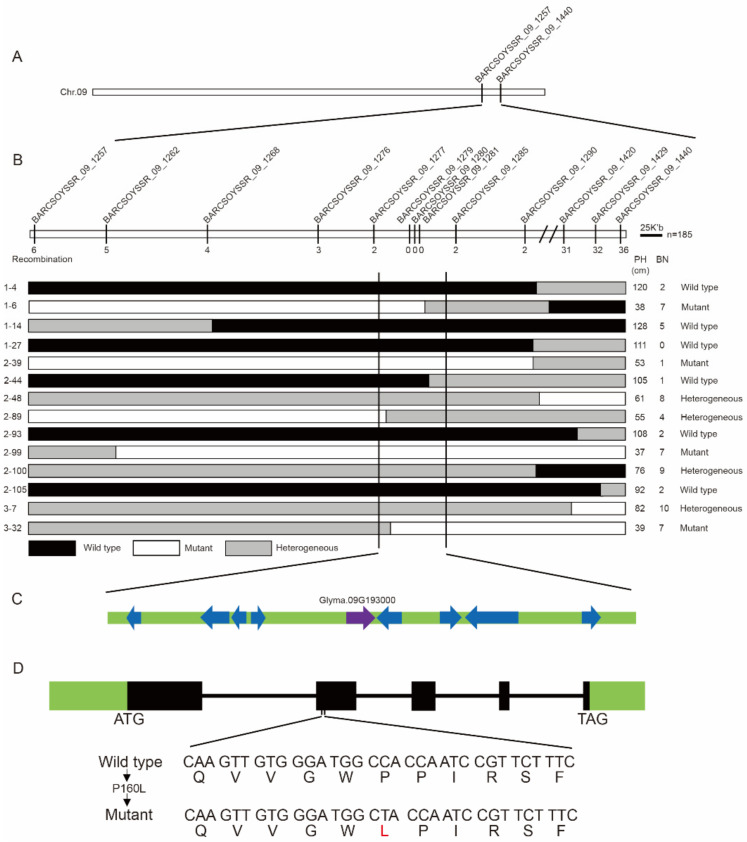
Map-based cloning of *GmIAA27* in soybean. (**A**,**B**) Initial mapping of *GmIAA27* using 185 plants from the F_2_ population derived from the cross *dmbn* × Dengke1. (**C**) Annotation in the candidate region according to the soybean cultivar Williams 82 reference genome. Arrowheads indicate putative genes predicted in Soybase (https://soybase.org/gb2/gbrowse/gmax2.0/; accessed on 22 February 2020). (**D**) Structure of the *GmIAA27* gene (*Glyma.09G193000*). A single-nucleotide substitution (C–T) was identified in the second exon of the *Glyma.09G193000* gene that converts a conserved Pro_160_ to Leu.

**Figure 4 ijms-23-08643-f004:**
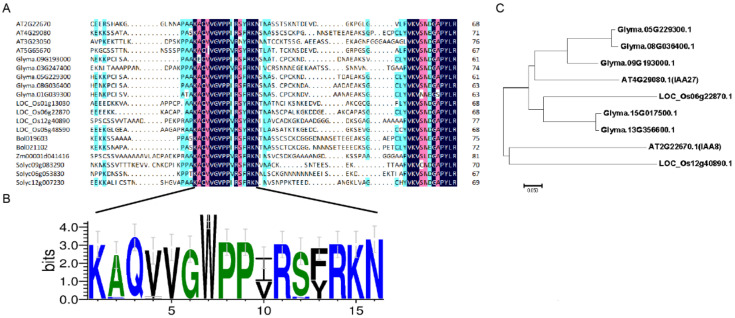
Sequence alignment and phylogenetic analysis of *GmIAA27* with homologue genes. (**A**) Alignment of partial amino acid sequence surrounding the 160th amino acid of Aux/IAA proteins from several plant species. (**B**) Conserved-sequence analysis of the partial amino acid sequence surrounding the 160th amino acid of Aux/IAA proteins from several plant species. (**C**) Phylogenetic tree of *Glyma.09G193000* and its gene homologs in soybean, *Arabidopsis*, and rice.

**Figure 5 ijms-23-08643-f005:**
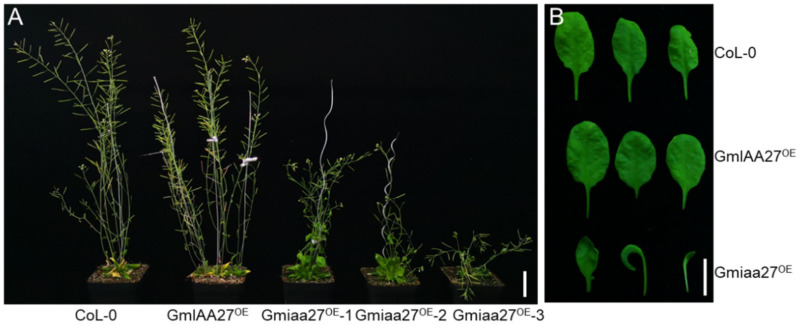
The functional confirmation of *Gmiaa27* in *Arabidopsis*. (**A**) Plant morphology of *Gmiaa27* transgenic plants at maturity; bar, 4 cm. (**B**) Leaf morphology of *Gmiaa27* transgenic plants at maturity; bar, 1 cm.

**Figure 6 ijms-23-08643-f006:**
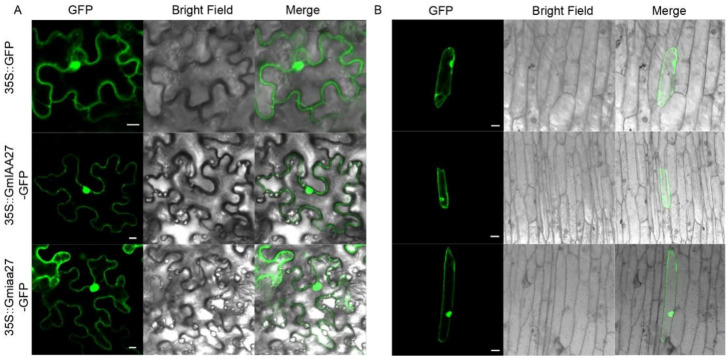
Subcellular localization of GmIAA27. (**A**) Subcellular location of 35S::GFP and 35S::GmIAA27/Gmiaa27-GFP in tobacco leaf cells. Bar, 10 μm. (**B**) Subcellular location of 35S::GFP and 35S::GmIAA27/Gmiaa27-GFP in onion epidermal cells. Bar, 10 μm.

**Figure 7 ijms-23-08643-f007:**
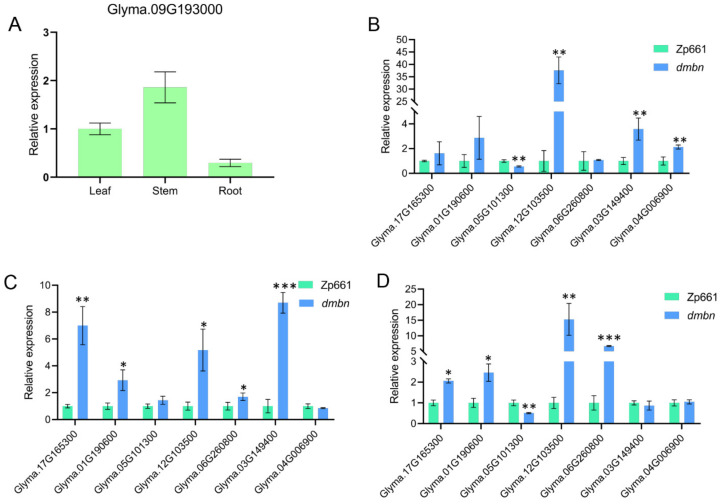
Relative expression analysis of GmIAA27 and auxin-responsive genes. (**A**) qRT-PCR relative expression of GmIAA27 in root, stem, and leaf. (**B**–**D**) qRT-PCR analysis of relative expression of GH3s in leaf (**B**), stem (**C**), and root (**D**) in Zp661 and *dmbn*. Values are means (± s.e.m.) (*n* = 3 plants each with three technical repeats), Student’s *t*-test was used for comparisons (* *p* ≤ 0.05, ** *p* ≤ 0.01, *** *p* ≤ 0.001).

**Figure 8 ijms-23-08643-f008:**
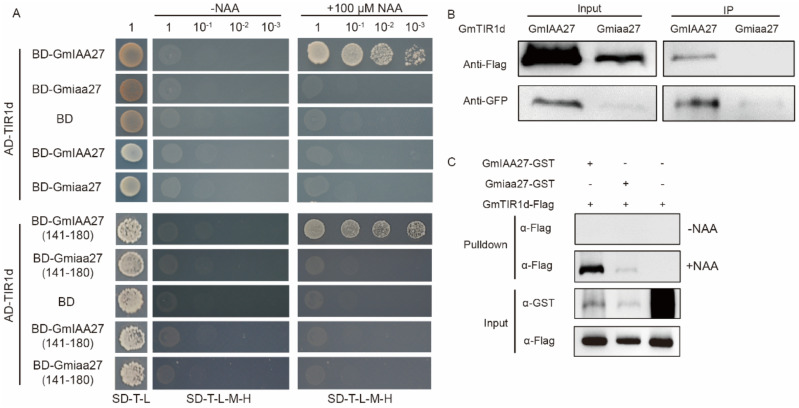
In vitro and in vivo interaction between GmIAA27/Gmiaa27 and GmTIR1d. (**A**) GmIAA27 and domain II of GmIAA27 (141-180) interacts with TIR1d in yeast cells with 100 μM NAA. SD/-T-L, selective medium lacking Trp and Leu; SD/-T-L-H-A, selective medium lacking Trp, Leu, His, and Ade. (**B**) Co-immunoprecipitation assays showing that GmIAA27 interacts with TIR1d in plant tobacco cells. (**C**) Pull-down assays showing that GmIAA27 interacts with TIR1d with 100 μM NAA.

## Data Availability

Not applicable.

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
