# Peer review of "GmIAA27 Encodes an AUX/IAA Protein Involved in Dwarfing and Multi-Branching in Soybean"

_ijms, 2022, doi:10.3390/ijms23158643_

Round 1

Reviewer 1 Report

The presented article is actual and informative, especially for the development of our understanding of dwarfism development mechanisms. A new gene encoding soybean dwarf phenotype was identified and evaluated using different methods including phenotyping, cytological analyses, hybridization, mapping with BSA-seq, Arabidpsis transformation and so on. Also the mechanism of this gene regulation was investigated. Finally it was found that new gene dmbn blocks auxin signaling thus affecting plant growth and brunching.

Article can be published after some English language editing.

Author Response

The presented article is actual and informative, especially for the development of our understanding of dwarfism development mechanisms. A new gene encoding soybean dwarf phenotype was identified and evaluated using different methods including phenotyping, cytological analyses, hybridization, mapping with BSA-seq, Arabidpsis transformation and so on. Also the mechanism of this gene regulation was investigated. Finally it was found that new gene dmbn blocks auxin signaling thus affecting plant growth and brunching.

Article can be published after some English language editing.

Reply: Thank you for the valuable comments.

Reviewer 2 Report

This manuscript first screened soybean dmbn mutant that shows shorter and more branched phenotype by EMS mutagenesis, mapped a responsible gene, GmIAA27, and characterized the gene function. The mapping strategy looks very fine and of high quality. However, here, I raised some concerns to be addressed or corrected before publication as below.

Fig. 1A. It is kind to show where is position 160 of GmIAA27.

Fig. 5. Arabidopsis originally possesses a full functional IAA27 gene. Under the background, ectopic overexpression of the mutated Gmiaa27 causes some morphological defects. This indicate that Gmiaa27 could have any artificial effects on development and/or negatively compete with AtIAA27. Authors should show any experimental evidences to why the ectopic expression caused the defects. Or, authors should delete Fig. 5 and the description.

Fig. 6. Intracellular localization of GmIAA27-GFP and Gmiaa27-GFP looks similar to that of free GFP. This data might not show real localization of IAA27. In addition, authors should try to examine intracellular localization of GFP-IAA27.

Fig. 7. GmIAA27 and Gmiaa27 have the same promoter but different mRNA accumulation. Is this because the mutation in Gmiaa27 destabilizes the mRNA, is there some kind of feedback, or is some other unexpected mutation affecting the mRNA? If other mutations are present, it is not possible to determine if the dmbn phenotype is solely dependent on Gmiaa27, authors should show any data or comments that can rule out the contribution of other mutations.

Fig. 8. The result of Fig. 8A is so doubtful, because I don’t know whether auxin is functional or not in yeasts and think there is a scientific discrepancy. So, this data may be dispensable.

Besides, in Fig. 8B and C, I suspect that the Gmiaa27 mutant protein is more unstable than GmIAA27 and does not appear to coprecipitate with GmTIR1d. If this scheme that GmIAA27 interacts with GmTIR1d is not properly demonstrated, the claims of this paper would be considered quite suspect.

Author Response

This manuscript first screened soybean dmbn mutant that shows shorter and more branched phenotype by EMS mutagenesis, mapped a responsible gene, GmIAA27, and characterized the gene function. The mapping strategy looks very fine and of high quality. However, here, I raised some concerns to be addressed or corrected before publication as below.

Fig. 1A. It is kind to show where is position 160 of GmIAA27.

Reply: Thank you for the comments. Position 160 of GmIAA27 has been showed in Fig. 3D according to your suggestion.

Fig. 5. Arabidopsis originally possesses a full functional IAA27 gene. Under the background, ectopic overexpression of the mutated Gmiaa27 causes some morphological defects. This indicate that Gmiaa27 could have any artificial effects on development and/or negatively compete with AtIAA27. Authors should show any experimental evidences to why the ectopic expression caused the defects. Or, authors should delete Fig. 5 and the description.

Reply: Thank you for the valuable comments. Ectopic overexpression of genes from other species in Arabidopsis has been a general method to study the function since Arabidopsis is easy for transformation. Just like most IAA family genes in rice, Arabidopsis, and Brassica napus, Gmiaa27 is also a gain-of-function mutation in our study. Therefore, mutants with corresponding phenotypes would be generated after ectopic expression or ontology expression. In Brassica napus, BnaC05.iaa7 driven by its native promoter to transform Arabidopsis displayed curly and wrinkled leaves at the seedling stage and an obvious dwarf stature without apical dominance at maturity. In this study, the mutant Gmiaa27 was transferred into WT Arabidopsis, which showed same phenotypic including deformed leaf and without apical dominance, which fully indicated that the GmIAA27 gene directly affected the growth and devel-opment of the plants., We have added some discussion about ectopic overexpression in Arabidopsis to make it more clear.

Fig. 6. Intracellular localization of GmIAA27-GFP and Gmiaa27-GFP looks similar to that of free GFP. This data might not show real localization of IAA27. In addition, authors should try to examine intracellular localization of GFP-IAA27.

Reply: Thank you for the valuable comments. GFP linked to the C-terminal of Aux/IAA proteins is a general method to reveal the localization of these proteins. Aux/IAA proteins share a conserved domain structure and are usually localized in the nucleus while some AUX/IAA are localized in both the nucleus and the cytoplasm. For example, AtIAA26, AtIAA17, ZmIAA2, ZmIAA11 and ZmIAA15 are confined to the nucleus and ZmIAA20, ZmIAA33 and AtIAA8 are localized in both the nucleus and the cytoplasm. Therefore, GmIAA27 might not act exclusively on transcriptional regulation in the nucleus, but may also play a role in cytoplasmic processes via an unknown mechanism. We have added discussion about Subcellular localization of IAA and will further investigated it by examining intracellular localization of GFP-IAA27.

Fig. 7. GmIAA27 and Gmiaa27 have the same promoter but different mRNA accumulation. Is this because the mutation in Gmiaa27 destabilizes the mRNA, is there some kind of feedback, or is some other unexpected mutation affecting the mRNA? If other mutations are present, it is not possible to determine if the dmbn phenotype is solely dependent on Gmiaa27, authors should show any data or comments that can rule out the contribution of other mutations.

Reply: Thank you for the valuable comments. In some mutants with gene domain variation, the mutation of target gene will also lead to the change of other genes, including the change of expression level, and the change caused by the gene structure itself is often the direct cause of the phenotypic mutation. In this study, the difference in GmIAA27 gene expression between wild-type and mutant could not fully explain the direct association with phenotype. We have deleted this description to make the expression more accurate,

Fig. 8. The result of Fig. 8A is so doubtful, because I don’t know whether auxin is functional or not in yeasts and think there is a scientific discrepancy. So, this data may be dispensable. Besides, in Fig. 8B and C, I suspect that the Gmiaa27 mutant protein is more unstable than GmIAA27 and does not appear to coprecipitate with GmTIR1d. If this scheme that GmIAA27 interacts with GmTIR1d is not properly demonstrated, the claims of this paper would be considered quite suspect.

Reply: Thank you for the valuable comments. In the relevant studies on the interaction between IAA and TIR1 proteins, it is necessary to verify whether exogenous auxin is added in the yeast two hybrid experiment [1-4]. Each co-receptor combination (IAA and TIR1) was evaluated on medium supplemented with increasing concentrations of auxin. Among the Arabidopsis Aux/IAAs tested, only IAA7 interacts with TIR1/AFBs in the absence of auxin. IAA5, IAA7 and IAA8 interact with all of the TIR1/AFBs at 0.1 μM IAA. IAA3 also bound TIR1, AFB1 and AFB2 at this concentration but was a poor substrate for AFB5. In contrast, IAA12, IAA28 and IAA29 required much higher concentrations of IAA to interact with the F-box proteins. IAA12 interacted specifically with TIR1 and AFB2 at 100 μM IAA, suggesting that, at least in the yeast system, higher concentration of IAA is required to form stable TIR1– or AFB2–IAA12 complexes [5]. Therefore, it is appropriate to use yeast two hybrid method to study GmIAA27 and TIR1 proteins.

In a study about 《The F-box protein TIR1 is an auxin receptor》, the researchers used an in vitro pull-down assay demonstrated that auxin promotes the Aux/IAA–SCFTIR1 interaction by binding directly to SCFTIR1[6]. In our study, we got the similar results。

In vivo protein interaction between IAA and TIR1 in rice was also carried out to determine whether OsTIR1 and OsAFB2–5 interact with Aux/IAA proteins. Co-immunoprecipitation (Co-IP) experiments was performed with OsIAA1 and OsIAA11 proteins in protoplasts using anti-FLAG antibodies, the result is that each of the OsTIR1 and OsAFB2–5 proteins interacted with OsIAA1 and OsIAA11. In our study, we found that expression of Gmiaa27 always resulted in decreased protein accumulation in tobacco, this phenomenon still needs further study. The IP samples of GmIAA27 (GFP-Trap was used to immuno-precipitate the GmIAA27 complexes) and TIR1d-Flag mixture, we use anti-Flag antibody to obtain target band of TIR1d-Flag, indicating that GmIAA27 interacts with TIR1d in vivo in tobacco leaves. Although the expression level of Gmiaa27 was low in tobacco leaves, the Co-IP interaction results similar to pulldown and Y2H were also detected after enrichment of Gmiaa27 protein. So, we speculate that the replacement of the first P by L in the degron motif GWPPV of GmIAA27 led directly to the weakening or absent the interaction between Gmiaa27 and TIR1d even in the presence of auxin.

[1] Arase F, Nishitani H, Egusa M, et al. IAA8 involved in lateral root formation interacts with the TIR1 auxin receptor and ARF transcription factors in Arabidopsis. PLoS One. 2012;7(8):e43414.

[2] Li H, Li J, Song J, et al. An auxin signaling gene BnaA3.IAA7 contributes to improved plant architecture and yield heterosis in rapeseed. New Phytol. 2019;222(2):837-851.

[3] Zhao B, Wang B, Li Z, et al. Identification and characterization of a new dwarf locus DS-4 encoding an Aux/IAA7 protein in Brassica napus. Theor Appl Genet. 2019;132(5):1435-1449.

[4] Zheng M, Hu M, Yang H, et al. Three BnaIAA7 homologs are involved in auxin/brassinosteroid-mediated plant morphogenesis in rapeseed (Brassica napus L.). Plant Cell Rep. 2019;38(8):883-897.

[5] Calderón Villalobos LI, Lee S, De Oliveira C, et al. A combinatorial TIR1/AFB-Aux/IAA co-receptor system for differential sensing of auxin. Nat Chem Biol. 2012;8(5):477-485.

[6] Dharmasiri N, Dharmasiri S, Estelle M. The F-box protein TIR1 is an auxin receptor. Nature. 2005;435(7041):441-445.

Reviewer 3 Report

The manuscript entitled “ GmIAA27 encodes an AUX/IAA protein involved in dwarfing and multi-branching in soybean” by Su et al aimed to study the mutant dmbn in soybean that was treated by EMS. 

The authors found that the mutant was shorter and more branched than WT. Further sequence analysis showed that Glyma.09g193000 encoding (GmIAA27) was mutated from C to T in the coding region, leading to the amino acid substitution of proline to leucine. More importantly, the authors studied the overexpression of the same type of mutant gene in Arabidopsis thaliana inhibit apical dominance and promoted branch development, which confirmed the hypothesis that such mutation controls the plant height and branch development. These results provide insights into the soybean plant development. 

Overall, the method used in the study is thorough. Conclusions are appropriate, and supported by the data. Statistical analysis is provided within the manuscript. The whole study is sound, and I recommend accepting it.

Author Response

The authors found that the mutant was shorter and more branched than WT. Further sequence analysis showed that Glyma.09g193000 encoding (GmIAA27) was mutated from C to T in the coding region, leading to the amino acid substitution of proline to leucine. More importantly, the authors studied the overexpression of the same type of mutant gene in Arabidopsis thaliana inhibit apical dominance and promoted branch development, which confirmed the hypothesis that such mutation controls the plant height and branch development. These results provide insights into the soybean plant development.

Overall, the method used in the study is thorough. Conclusions are appropriate, and supported by the data. Statistical analysis is provided within the manuscript. The whole study is sound, and I recommend accepting it.

Reply: Thank you for the valuable comments.

Round 2

Reviewer 2 Report

I have understood auxin's functionality in yeast system. 

Although little bit suspiciousness looks remain in IP assay (Fig. 8b), interaction between GmTIR1d and GmIAA27 but not Gmiaa27 under NAA is reliable (Fig. 8a and 8c).